# Improved Precision and Recall Metric for Assessing Generative Models

**Tuomas Kynkäänniemi**[*]
Aalto University
NVIDIA
tuomas.kynkaanniemi@aalto.fi

**Tero Karras**
NVIDIA
tkarras@nvidia.com

**Samuli Laine**
NVIDIA
slaine@nvidia.com

**Jaakko Lehtinen**
Aalto University
NVIDIA
jlehtinen@nvidia.com

**Timo Aila**
NVIDIA
taila@nvidia.com

## Abstract

The ability to automatically estimate the quality and coverage of the samples produced by a generative model is a vital requirement for driving algorithm research. We present an evaluation metric that can separately and reliably measure both of these aspects in image generation tasks by forming explicit, non-parametric representations of the manifolds of real and generated data. We demonstrate the effectiveness of our metric in StyleGAN and BigGAN by providing several illustrative examples where existing metrics yield uninformative or contradictory results. Furthermore, we analyze multiple design variants of StyleGAN to better understand the relationships between the model architecture, training methods, and the properties of the resulting sample distribution. In the process, we identify new variants that improve the state-of-the-art. We also perform the first principled analysis of truncation methods and identify an improved method. Finally, we extend our metric to estimate the perceptual quality of individual samples, and use this to study latent space interpolations.

## 1 Introduction

The goal of generative methods is to learn the manifold of the training data so that we can subsequently generate novel samples that are indistinguishable from the training set. While the quality of results from generative adversarial networks (GAN) [7], variational autoencoders (VAE) [14], autoregressive models [29, 30], and likelihood-based models [6, 13] have seen rapid improvement recently [11, 8, 28, 20, 4, 12], the automatic evaluation of these results continues to be challenging.

When modeling a complex manifold for sampling purposes, two separate goals emerge: individual samples drawn from the model should be faithful to the examples (they should be of "high quality"), and their variation should match that observed in the training set. The most widely used metrics, such as Fréchet Inception Distance (FID) [9], Inception Score (IS) [25], and Kernel Inception Distance (KID) [2], group these two aspects to a single value without a clear tradeoff. We illustrate by examples that this makes diagnosis of model performance difficult. For instance, it is interesting that while recent state-of-the-art generative methods [4, 13, 12] claim to optimize FID, in the end the (uncurated) results are almost always produced using another model that explicitly sacrifices variation, and often FID, in favor of higher quality samples from a truncated subset of the domain [17].

---

[*]This work was done during an internship at NVIDIA.

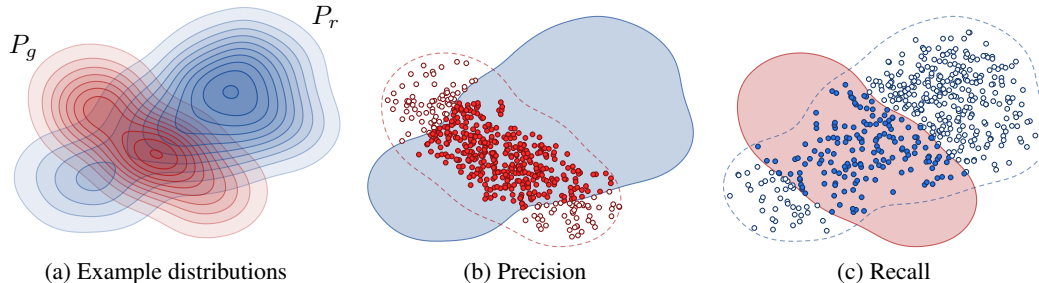

| (a) Example distributions | (b) Precision | (c) Recall |

Figure 1: Definition of precision and recall for distributions [24]. (a) Denote the distribution of real images with $P_r$ (blue) and the distribution of generated images with $P_g$ (red). (b) Precision is the probability that a random image from $P_g$ falls within the support of $P_r$. (c) Recall is the probability that a random image from $P_r$ falls within the support of $P_g$.

Meanwhile, insufficient coverage of the underlying manifold continues to be a challenge for GANs. Various improvements to network architectures and training procedures tackle this issue directly [25, 19, 11, 15]. While metrics have been proposed to estimate the degree of variation, these have not seen widespread use as they are subjective [1], domain specific [19], or not reliable enough [23].

Recently, Sajjadi et al. [24] proposed a novel metric that expresses the quality of the generated samples using two separate components: *precision* and *recall*. Informally, these correspond to the average sample quality and the coverage of the sample distribution, respectively. We discuss their metric (Section 1.1) and characterize its weaknesses that we later demonstrate experimentally. Our primary contribution is an improved precision and recall metric (Section 2) which provides explicit visibility of the tradeoff between sample quality and variety. Source code of our metric is available at `https://github.com/kynkaat/improved-precision-and-recall-metric`.

We demonstrate the effectiveness of our metric using two recent generative models (Section 3), StyleGAN [12] and BigGAN [4]. We then use our metric to analyze several variants of StyleGAN (Section 4) to better understand the design decisions that determine result quality, and identify new variants that improve the state-of-the-art. We also perform the first principled analysis of truncation methods [17, 13, 4, 12]. Finally, we extend our metric to estimate the quality of individual generated samples (Section 5), offering a way to measure the quality of latent space interpolations.

## 1.1 Background

Sajjadi et al. [24] introduce the classic concepts of precision and recall to the study of generative models, motivated by the observation that FID and related density metrics cannot be used for making conclusions about precision and recall: a low FID may indicate high precision (realistic images), high recall (large amount of variation), or anything in between. We share this motivation.

From the classic viewpoint, *precision* denotes the fraction of generated images that are realistic, and *recall* measures the fraction of the training data manifold covered by the generator (Figure 1). Both are computed as expectations of binary set membership over a distribution, i.e., by measuring how likely is it that an image drawn from one distribution is classified as falling under the support of the other distribution. In contrast, Sajjadi et al. [24] formulate precision and recall through the relative probability densities of the two distributions. The choice of modeling the relative densities comes from an ambiguity, i.e., should the differences between the two distributions be attributed to the generator covering the real distribution inadequately or is the generator producing samples that are unrealistic. The authors resolve this ambiguity by modeling a continuum of precision/recall values where the extrema correspond to the classic definitions. In addition to raising the question of which value to use, their practical algorithm cannot reliably estimate the extrema due to its reliance on relative densities: it cannot, for instance, correctly interpret situations where large numbers of samples are packed together, e.g., as a result of mode collapse or truncation. The $k$-nearest neighbors based two-sample test by Lopez-Paz et. al. [16] suffers from the same problem. Parallel with our work, Simon et al. [26] extend Sajjadi's formulation to arbitrary probability distributions and provide a practical algorithm that estimates precision and recall by training a post hoc classifier.

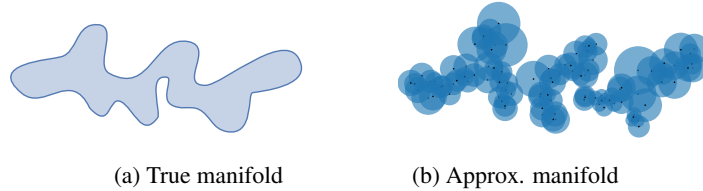

(a) True manifold          (b) Approx. manifold

Figure 2: (a) An example manifold in a feature space. (b) Estimate of the manifold obtained by sampling a set of points and surrounding each with a hypersphere that reaches its $k$th nearest neighbor.

We argue that the classic definition of precision and recall is sufficient for disentangling the effects of sample quality and manifold coverage. This can be partially justified by observing that precision and recall correspond to the vertical and horizontal extremal cases in Lin et al.'s [15] theoretically founded analysis of mode collapse regions. In order to approximate these quantities directly, we construct adaptive-resolution finite approximations to the real and generated manifolds that are able to answer binary membership queries: *"does sample $x$ lie in the support of distribution $P$?"*. Together with existing density-based metrics, such as FID, our precision and recall scores paint a highly informative picture of the distributions produced by generative image models. In particular, they make effects in the "null space" of FID clearly visible.

## 2  Improved precision and recall metric using $k$-nearest neighbors

We will now describe our improved precision and recall metric that does not suffer from the weaknesses listed in Section 1.1. The key idea is to form explicit non-parametric representations of the manifolds of real and generated data, from which precision and recall can be estimated.

Similar to Sajjadi et al. [24], we draw real and generated samples from $X_r \sim P_r$ and $X_g \sim P_g$, respectively, and embed them into a high-dimensional feature space using a pre-trained classifier network. We denote feature vectors of the real and generated images by $\phi_r$ and $\phi_g$, respectively, and the corresponding sets of feature vectors by $\mathbf{\Phi}_r$ and $\mathbf{\Phi}_g$. We take an equal number of samples from each distribution, i.e., $|\mathbf{\Phi}_r| = |\mathbf{\Phi}_g|$.

For each set of feature vectors $\mathbf{\Phi} \in \{\mathbf{\Phi}_r, \mathbf{\Phi}_g\}$, we estimate the corresponding manifold in the feature space as illustrated in Figure 2. We obtain the estimate by calculating pairwise Euclidean distances between all feature vectors in the set and, for each feature vector, forming a hypersphere with radius equal to the distance to its $k$th nearest neighbor. Together, these hyperspheres define a volume in the feature space that serves as an estimate of the true manifold. To determine whether a given sample $\phi$ is located within this volume, we define a binary function

$$f(\phi, \mathbf{\Phi}) = \begin{cases} 1, & \text{if } \left\| \phi - \phi' \right\|_2 \leq \left\| \phi' - \mathrm{NN}_k\left(\phi', \mathbf{\Phi}\right) \right\|_2 \text{ for at least one } \phi' \in \mathbf{\Phi} \\ 0, & \text{otherwise}, \end{cases} \tag{1}$$

where $\mathrm{NN}_k\left(\phi', \mathbf{\Phi}\right)$ returns $k$th nearest feature vector of $\phi'$ from set $\mathbf{\Phi}$. In essence, $f(\phi, \mathbf{\Phi}_r)$ provides a way to determine whether a given image looks realistic, whereas $f(\phi, \mathbf{\Phi}_g)$ provides a way to determine whether it could be reproduced by the generator. We can now define our metric as

$$\mathrm{precision}(\mathbf{\Phi}_r, \mathbf{\Phi}_g) = \frac{1}{|\mathbf{\Phi}_g|} \sum_{\phi_g \in \mathbf{\Phi}_g} f(\phi_g, \mathbf{\Phi}_r) \quad \mathrm{recall}(\mathbf{\Phi}_r, \mathbf{\Phi}_g) = \frac{1}{|\mathbf{\Phi}_r|} \sum_{\phi_r \in \mathbf{\Phi}_r} f(\phi_r, \mathbf{\Phi}_g) \tag{2}$$

In Equation (2), precision is quantified by querying for each generated image whether the image is within the estimated manifold of real images. Symmetrically, recall is calculated by querying for each real image whether the image is within estimated manifold of generated images. See Appendix A in the supplement for pseudocode.

In practice, we compute the feature vector $\phi$ for a given image by feeding it to a pre-trained VGG-16 classifier [27] and extracting the corresponding activation vector after the second fully connected layer. Brock et al. [4] show that the nearest neighbors in this feature space are meaningful in the sense that they correspond to semantically similar images. Meanwhile, Zhang et al. [32] use the intermediate activations of multiple convolutional layers of VGG-16 to define a perceptual metric, which they

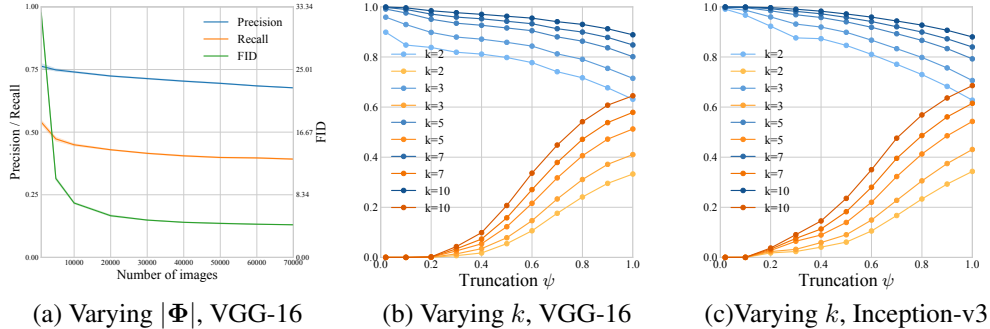

(a) Varying $|\mathbf{\Phi}|$, VGG-16    (b) Varying $k$, VGG-16    (c) Varying $k$, Inception-v3

Figure 3: (a) Our metric behaves similarly to FID in terms of varying sample count. (b) Precision (blue) and recall (orange) for several neighborhood sizes $k$. Larger $k$ increases both numbers. Here a trained model ($\psi = 1$) was expanded to a family of models by artificially limiting the variation in the results. We would expect the precision and recall to reach 1.0 and 0.0, respectively, when $\psi \to 0$. (c) Using Inception-v3 features instead of VGG-16 yields a substantially similar result.

show to correlates well with human judgment for image corruptions. We have tested both approaches and found that feature space, used by Brock at al., works considerably better for the purposes of our metric, presumably because it places less emphasis on the exact spatial arrangement — sparsely sampled manifolds rarely include near-exact matches in terms of spatial structure.

Like FID, our metric is weakly affected by the number of samples taken (Figure 3a). Since it is standard practice to quote FIDs with 50k samples, we adopt the same design point for our metric as well. The size of the neighborhood, $k$, is a compromise between covering the entire manifold (large values) and overestimating its volume as little as possible (small values). In practice, we have found that higher values of $k$ increase the precision and recall estimates in a fairly consistent fashion, and lower values of $k$ decrease them, until they start saturating at 1.0 or 0.0 (Figure 3b). Tests with various datasets and GANs showed that $k = 3$ is a robust choice that avoids saturating the values most of the time. Thus we use $k = 3$ and $|\mathbf{\Phi}| = 50000$ in all our experiments unless stated otherwise. Figure 3c further shows that the qualitative behavior of our metric is not limited to VGG-16 – which we use in all tests – as Inception-v3 features lead to very similar results.

## 3 Precision and recall of state-of-the-art generative models

In this section, we demonstrate that precision and recall computed using our method correlate well with the perceived quality and variation of generated distributions, and compare our metric with Sajjadi et al.'s method [24] as well as the widely used FID metric [9]. For Sajjadi et al.'s method, we use 20 clusters and report $F_{1/8}$ and $F_8$ as proxies for precision and recall, respectively, as recommended by the authors. We examine two state-of-the-art generative models, StyleGAN [12] trained with the FFHQ dataset, and BigGAN [4] trained on ImageNet [5].

**StyleGAN** Figure 4 shows the results of various metrics in four StyleGAN setups. These setups exhibit different amounts of truncation and training time, and have been selected to illustrate how the metrics behave with varying output image distributions. Setup A is heavily truncated, and the generated images are of high quality but very similar to each other in terms of color, pose, background, etc. This leads to high precision and low recall, as one would expect. Moving to setup B increases variation, which improves recall, while the image quality and thus precision is somewhat compromised. Setup C is the FID-optimized configuration in [12]. It has even more variation in terms of color schemes and accessories such as hats and sunglasses, further improving recall. However, some of the faces start to become distorted which reduces precision. Finally, setup D preserves variation and recall, but nearly all of the generated images have low quality, indicated by much lower precision as expected.

In contrast, the method of Sajjadi et al. [24] indicates that setups B, C and D are all essentially perfect, and incorrectly assigns setup A the lowest precision. Looking at FID, setups B and D appear almost equally good, illustrating how much weight FID places on variation compared to image quality, also evidenced by the high FID of setup A. Setup C is ranked as clearly the best by FID despite the obvious

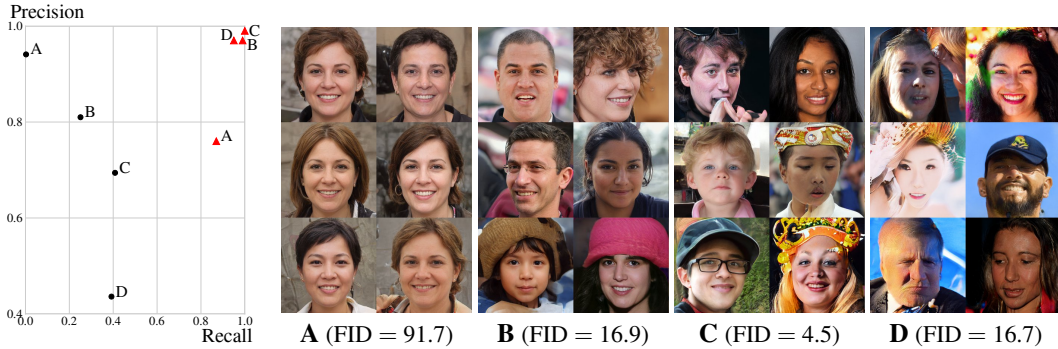

| **A** (FID = 91.7) | **B** (FID = 16.9) | **C** (FID = 4.5) | **D** (FID = 16.7) |

Figure 4: Comparison of our method (black dots), Sajjadi et al.'s method [24] (red triangles), and FID for 4 StyleGAN setups. We recommend zooming in to better assess the quality of images.

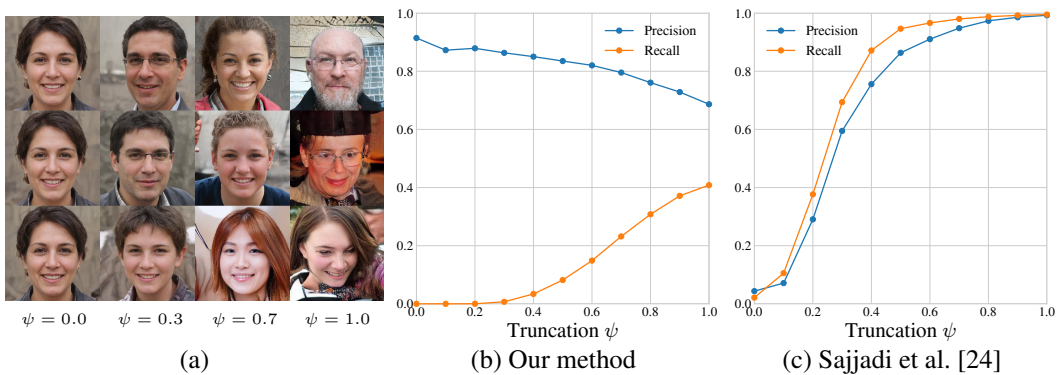

|  (a) | (b) Our method | (c) Sajjadi et al. [24] |

Figure 5: (a) Example images produced by StyleGAN [12] trained using the FFHQ dataset. It is generally agreed [17, 4, 13, 12] that truncation provides a tradeoff between perceptual quality and variation. (b) With our method, the maximally truncated setup ($\psi = 0$) has zero recall but high precision. As truncation is gradually removed, precision drops and recall increases as expected. The final recall value approximates the fraction of training set the generator can reproduce (generally well below 100%). (c) The method of Sajjadi et al. reports both precision and recall increasing as truncation is removed, contrary to the expected behavior, and the final numerical values of both precision and recall seem excessively high.

image artifacts. The ideal tradeoff between quality and variation depends on the intended application, but it is unclear which application might favor setup D where practically all images are broken over setup B that produces high-quality samples at a lower variation. Our metric provides explicit visibility on this tradeoff and allows quantifying the suitability of a given model for a particular application.

Figure 5 applies gradually stronger truncation [17, 4, 13, 12] on precision and recall using a single StyleGAN generator. Our method again works as expected, while the method of Sajjadi et al. does not. We hypothesize that their difficulties are a result of truncation packing a large number of generated images into a small region in the embedding space. This may result in clusters that contain no real images in that region, and ultimately causes the metric to incorrectly report low precision. The tendency to underestimate precision can be alleviated by using fewer clusters, but doing so leads to overestimation of recall. Our metric does not suffer from this problem because the manifolds of real and generated images are estimated separately, and the distributions are never mixed together.

**BigGAN**    Brock et al. recently presented BigGAN [4], a high-quality generative network able to synthesize images for ImageNet [5]. ImageNet is a diverse dataset containing 1000 classes with ~1300 training images for each class. Due to the large amount of variation within and between classes, generative modeling of ImageNet has proven to be a challenging problem [21, 31, 4]. Brock et al. [4] list several ImageNet classes that are particularly easy or difficult for their method. The difficult classes often contain precise global structure or unaligned human faces, or they are underrepresented

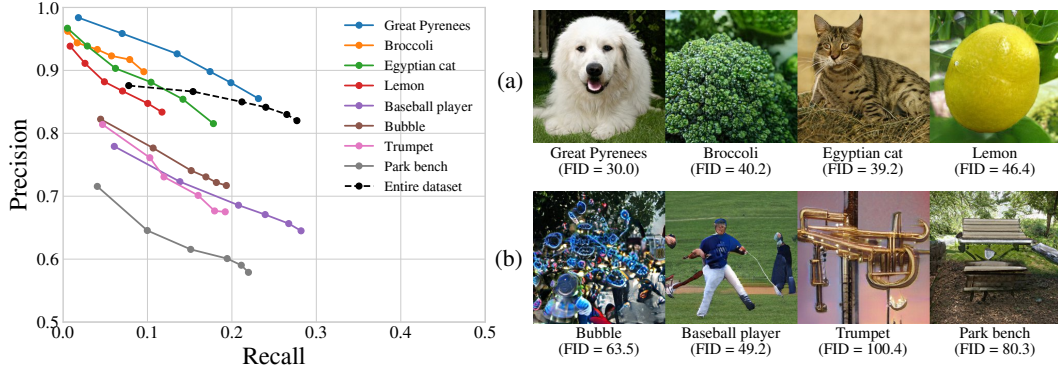

Figure 6: Our precision and recall for four easy (a) and four difficult (b) ImageNet classes using BigGAN. For each class we sweep the truncation parameter $\psi$ linearly from 0.3 to 1.0, left-to-right. The FIDs refer to a non-truncated model, i.e., $\psi = 1.0$. The per-class metrics were computed using all available training images of the class and an equal number of generated images, while the curve for the entire dataset was computed using 50k real and generated images.

in the dataset. The easy classes are largely textural, lack exact global structure, and are common in the dataset. Dogs are a noteworthy special case in ImageNet: with almost a hundred different dog breeds listed as separate classes, there is much more training data for dogs than for any other class, making them artificially easy. To a lesser extent, the same applies to cats that occupy $\sim$10 classes.

Figure 6 illustrates the precision and recall for some of these classes over a range of truncation values. We notice that precision is invariably high for the suspected easy classes, including cats and dogs, and clearly lower for the difficult ones. Brock et al. state that the quality of generated samples increases as more truncation is applied, and the precision as reported by our method is in line with this observation. Recall paints a more detailed picture. It is very low for classes such as "Lemon" or "Broccoli", implying much of the variation has been missed, but FID is nevertheless quite good for both. Since FID corresponds to a Wasserstein-2 distance in the feature space, low intrinsic variation implies low FID even when much of that variation is missed. Correspondingly, recall is clearly higher for the difficult classes. Based on visual inspection, these classes have a lot of intra-class variation that BigGAN training has successfully modeled. Dogs and cats show recall similar to the difficult classes, and their image quality and thus precision is likely boosted by the additional training data.

## 4 Using precision and recall to analyze and improve StyleGAN

Generative models have seen rapid improvements recently, and FID has risen as the de facto standard for determining whether a proposed technique is considered beneficial or not. However, as we have shown in Section 3, relying on FID alone may hide important qualitative differences in the results and it may inadvertently favor a particular tradeoff between precision and recall that is not necessarily aligned with the actual goals. In this section, we use our metric to shed light onto some of the design decisions associated with the model itself. Appendix C in the supplement performs a similar, principled analysis for truncation methods. We use StyleGAN [12] in all experiments, trained with FFHQ at $1024 \times 1024$.

### 4.1 Network architectures and training configurations

To avoid drawing false conclusions when comparing different training runs, we must properly account for the stochastic nature of the training process. For example, we have observed that FID can often vary by up to $\pm 14\%$ between consecutive training iterations with StyleGAN. The common approach is to amortize this variation by taking multiple snapshots of the model at regular intervals and selecting the best one for further analysis [12]. With our metric, however, we are faced with the problem of multiobjective optimization [3]: the snapshots represent a wide range of different tradeoffs between precision and recall, as illustrated in Figure 7a. To avoid making assumptions about the desired tradeoff, we identify the Pareto frontier, i.e., the minimal subset of snapshots that is guaranteed to contain the optimal choice for any given tradeoff.

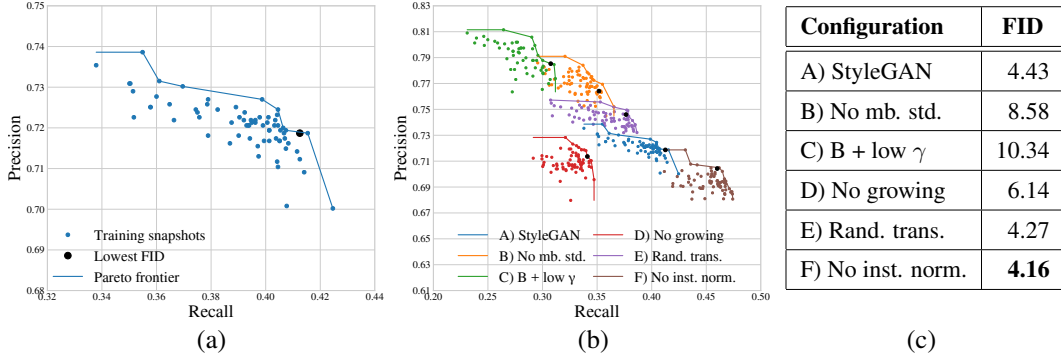

| Configuration | FID |
|---|---|
| A) StyleGAN | 4.43 |
| B) No mb. std. | 8.58 |
| C) B + low $\gamma$ | 10.34 |
| D) No growing | 6.14 |
| E) Rand. trans. | 4.27 |
| F) No inst. norm. | **4.16** |

(a)          (b)          (c)

Figure 7: (a) Precision and recall for different snapshots of StyleGAN taken during the training, along with their corresponding Pareto frontier. We use the standard training configuration by Karras et al. [12] with FFHQ and $\psi = 1$. (b) Different training configurations lead to vastly different tradeoffs between precision and recall. (c) Best FID obtained for each configuration (lower is better).

Figure 7b shows the Pareto frontiers for several variants of StyleGAN. The baseline configuration (A) has a dedicated *minibatch standard deviation* layer that aims to increase variation in the generated images [11, 15]. Using our metric, we can confirm that this is indeed the case: removing the layer shifts the tradeoff considerably in favor of precision over recall (B). We observe that $R_1$ regularization [18] has a similar effect: reducing the $\gamma$ parameter by $100\times$ shifts the balance even further (C). Karras et al. [11] argue that their progressive growing technique improves both quality and variation, and indeed, disabling it reduces both aspects (D). Moreover, we see that randomly translating the inputs of the discriminator by $-16\ldots16$ pixels improves precision (E), whereas disabling instance normalization in the AdaIN operation [10], unexpectedly, improves recall (F).

Figure 7c shows the best FID obtained for each configuration; the corresponding snapshots are highlighted in Figure 7a,b. We see that FID favors configurations with high recall (A, F) over the ones with high precision (B, C), and the same is also true for the individual snapshots. The best configuration in terms of recall (F) yields a new state-of-the-art FID for this dataset. Random translation (E) is an exceptional case: it improves precision at the cost of recall, similar to (B), but also manages to slightly improve FID at the same time. We leave an in-depth study of these effects for future work.

## 5 Estimating the quality of individual samples

While our precision metric provides a way to assess the overall quality of a population of generated images, it yields only a binary result for an individual sample and therefore is not suitable for ranking images by their quality. Here, we present an extension of the classification function $f$ (Equation 1) that provides a continuous estimate of how close a given sample is to the manifold of real images.

We define a *realism score* $R$ that increases the closer an image is to the manifold and decreases the further an image is from the manifold. Let $\phi_g$ be a feature vector of a generated image and $\phi_r$ a feature vector of a real image from set $\mathbf{\Phi}_r$. Realism score of $\phi_g$ is calculated as

$$R(\phi_g, \mathbf{\Phi}_r) = \max_{\phi_r} \left\{ \frac{\|\phi_r - \mathrm{NN}_k\left(\phi_r, \mathbf{\Phi}_r\right)\|_2}{\|\phi_g - \phi_r\|_2} \right\}. \tag{3}$$

This is a continuous extension of $f(\phi_g, \mathbf{\Phi}_r)$ with the simple relation that $f(\phi_g, \mathbf{\Phi}_r) = 1$ iff $R(\phi_g, \mathbf{\Phi}_r) \geq 1$. In other words, when $R \geq 1$, the feature vector $\phi_g$ is inside the ($k$-NN induced) hypersphere of at least one $\phi_r$.

With any finite training set, the $k$-NN hyperspheres become larger in regions where the training samples are sparse, i.e., regions with low representation. When measuring the quality of a large population of generated images, these underrepresented regions have little impact as it is unlikely that too many generated samples land there — even though the hyperspheres may be large, they are sparsely located and cover a small volume of space in total. However, when computing the realism

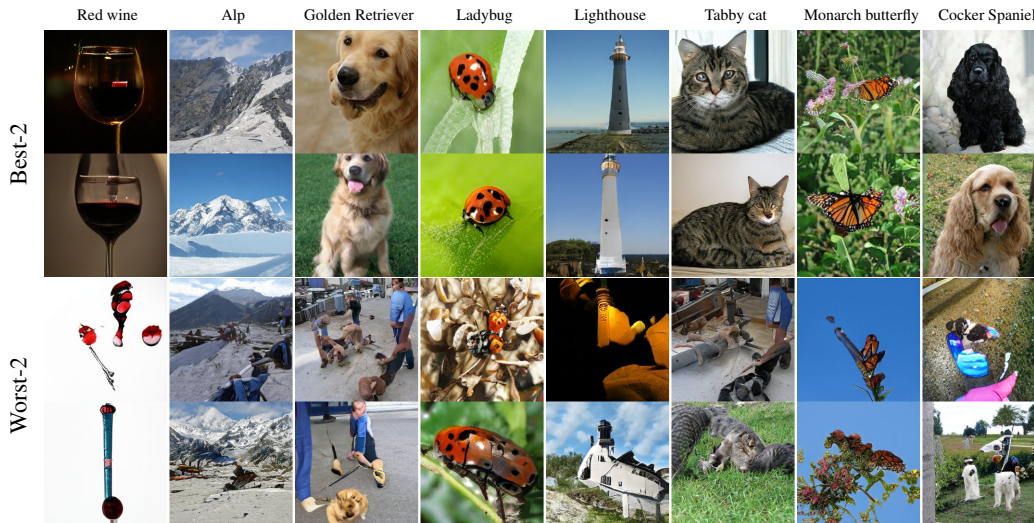

Figure 8: Quality of individual samples of BigGAN from eight classes. Top: Images with high realism. Bottom: Images with low realism. We show two images with the highest and lowest realism score selected from 1000 non-truncated images.

score for a single image, a sample that happens to land in such a fringe hypersphere may obtain a wildly inaccurate score. Large errors, even if they are rare, would undermine the usefulness of the metric. We tackle this problem by discarding half of the hyperspheres with the largest radii. In other words, the maximum in Equation 3 is not taken over all $\phi_r \in \Phi_r$ but only over those $\phi_r$ whose associated hypersphere is smaller than the median. This pruning yields an overconservative estimate of the real manifold, but it leads to more consistent realism scores. Note that we use this approach only with $R$, not with $f$.

Figure 8 shows example images from BigGAN with high and low realism. In general, the samples with high realism display a clear object from the given class, whereas the object is often distorted to unrecognizable for the low realism images. Appendix D in the supplement provides more examples.

## 5.1 Quality of interpolations

An interesting application for the realism score is to evaluate the quality of interpolations. We do this with StyleGAN using linear interpolation in the intermediate latent space $\mathcal{W}$ as suggested by Karras et al. [12]. Figure 9 shows four example interpolation paths with randomly sampled latent vectors as endpoints. Paths A appears to be located completely inside the real manifold, path D completely outside it, and paths B and C have one endpoint inside the real manifold and one outside it. The realism scores assigned to paths A–D correlate well with the perceived image quality: Images with low scores contain multiple artifacts and can be judged to be outside the real manifold, and vice versa for high-scoring images. See Appendix D in the supplement for additional examples.

We can use interpolations to investigate the shape of the subset of $\mathcal{W}$ that produces realistic-looking images. In this experiment, we sampled without truncation 1M latent vectors in $\mathcal{W}$ for which $R \geq 1$, giving rise to 500k interpolation paths with both endpoints on the real manifold. It would be unrealistic to expect all intermediate images on these paths to also have $R \geq 1$, so we chose to consider an interpolation path where more than 25% of the intermediate images have $R < 0.9$ as straying too far from the real manifold. Somewhat surprisingly, we found that only 2.4% of the paths crossed unrealistic parts of $\mathcal{W}$ under this definition, suggesting that the subset of $\mathcal{W}$ on the real manifold is highly convex. We see potential in using the realism score for measuring the shape of this region in $\mathcal{W}$ with greater accuracy, possibly allowing the exclusion of unrealistic images in a more refined manner than with truncation-like methods.

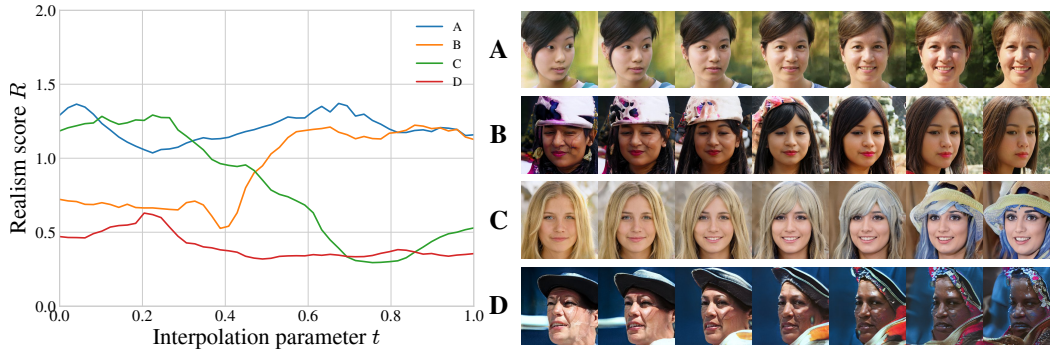

Figure 9: Realism score for four interpolation paths as function of linear interpolation parameter $t$ and corresponding images from paths A–D. We did not use truncation when generating the images.

# 6 Conclusion

We have demonstrated through several experiments that the separate assessment of precision and recall can reveal interesting insights about generative models and can help to improve them further. We believe that the separate quantification of precision can also be useful in the context of image-to-image translation [33], where the quality of individual images is of great interest.

Using our metric, we have identified previously unknown training configuration-related effects in Section 4.1, raising the question whether truncation is really necessary if similar tradeoffs can be achieved by modifying the training configuration appropriately. We leave the in-depth study of these effects for future work.

Finally, it has recently emerged that density models can be incapable of assessing whether a given example belongs to the training distribution [22]. By explicitly modeling the real manifold, our metrics may provide an alternative way for estimating this.

# 7 Acknowledgements

We thank David Luebke for helpful comments; Janne Hellsten, and Tero Kuosmanen for compute infrastructure.

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
