[Supplementary Material]

# Supplement: Improved Precision and Recall Metric for Assessing Generative Models

**Tuomas Kynkäänniemi**[*]
Aalto University
NVIDIA
tuomas.kynkaanniemi@aalto.fi

**Tero Karras**
NVIDIA
tkarras@nvidia.com

**Samuli Laine**
NVIDIA
slaine@nvidia.com

**Jaakko Lehtinen**
Aalto University
NVIDIA
jlehtinen@nvidia.com

**Timo Aila**
NVIDIA
taila@nvidia.com

## A  Pseudocode and implementation details

---
**Algorithm 1** $k$-NN precision and recall pseudocode.

---
**Input:** Set of real and generated images $(X_r, X_g)$, feature network $\mathcal{F}$, neighborhood size $k$.

1: **function** PRECISION-RECALL($X_r, X_g, \mathcal{F}, k$)
2:     $\mathbf{\Phi}_r \quad\quad \leftarrow \mathcal{F}(X_r)$
3:     $\mathbf{\Phi}_g \quad\quad \leftarrow \mathcal{F}(X_g)$
4:     precision $\leftarrow$ MANIFOLD-ESTIMATE($\mathbf{\Phi}_r, \mathbf{\Phi}_g, k$)
5:     recall $\quad\; \leftarrow$ MANIFOLD-ESTIMATE($\mathbf{\Phi}_g, \mathbf{\Phi}_r, k$)
6:     **return** precision, recall

7: **function** MANIFOLD-ESTIMATE($\mathbf{\Phi}_a, \mathbf{\Phi}_b, k$)
8:     Approximate manifold of $\mathbf{\Phi}_a$.
9:     **for** $\phi \in \mathbf{\Phi}_a$ **do**
10:        $\boldsymbol{d} \; \leftarrow \big\{ \big\| \phi - \phi' \big\|_2 \big\}$ **for** all $\phi' \in \mathbf{\Phi}_a$     ▷ Pairwise distances to all points in $\mathbf{\Phi}_a$.
11:        $r_\phi \leftarrow \min_{k+1}(\boldsymbol{d})$                    ▷ $(k+1)$-th smallest value to exclude $\phi$ itself.
12:     Compute how many points from $\mathbf{\Phi}_b$ are within the approximated manifold of $\mathbf{\Phi}_a$.
13:     $n \leftarrow 0$
14:     **for** $\phi \in \mathbf{\Phi}_b$ **do**
15:        **if** $\big\| \phi - \phi' \big\|_2 \le r_{\phi'}$ **for** any $\phi' \in \mathbf{\Phi}_a$ **then**
16:           $n \leftarrow n+1$
17:     **return** $n/|\mathbf{\Phi}_b|$

---

Algorithm 1 shows the pseudocode for our method. The main function PRECISION-RECALL evaluates precision and recall for given sets of real and generated images, $X_r$ and $X_g$, by embedding them in a feature space defined by $\mathcal{F}$ (lines 2–3) and estimating the corresponding manifolds using MANIFOLD-ESTIMATE (lines 4–6). The helper function MANIFOLD-ESTIMATE takes two sets of feature vectors $\mathbf{\Phi}_a, \mathbf{\Phi}_b$ as inputs. It forms an estimate for the manifold of $\mathbf{\Phi}_a$ and counts how many points from $\mathbf{\Phi}_b$ are located within the manifold. Estimating the manifold requires computing the pairwise distances between all feature vectors $\phi \in \mathbf{\Phi}_a$ and, for each $\phi$, tabulating the distance to its $k$-th nearest neighbor (lines 9–11). These distances are then used to determine the fraction of feature vectors

---

[*]This work was done during an internship at NVIDIA.

Figure 1: (a) Real data covers five modes (1–5) and the generated data is expanded, one mode at a time, to cover the real modes (1–5) and five extraneous modes (6–10). Both metrics were evaluated using 20k real and generated samples. (b) Results from our metric with $k = 3$. (c) Results from the method of Sajjadi et al. [6].

$\phi \in \mathbf{\Phi}_b$ that are located within the manifold (lines 13–17). Note that in the pseudocode feature vectors $\phi$ are processed one by one on lines 9 and 14 but in a practical implementation they can be processed in mini-batches to improve efficiency.

We use NVIDIA Tesla V100 GPU to run our implementation. A high-quality estimate using 50k images in both $X_r$ and $X_g$ takes $\sim 8$ minutes to run on a single GPU. For comparison, evaluating FID using the same data takes $\sim 4$ minutes and generating 50k images ($1024 \times 1024$) with StyleGAN using one GPU takes $\sim 14$ minutes. Our implementation can be found at `https://github.com/kynkaat/improved-precision-and-recall-metric`.

## B  Precision and recall with synthetic dataset

In Figure 1 we replicate the mode dropping and invention experiment from [6], albeit with a 10-class 2D Gaussian mixture model instead of CIFAR-10 images. As in [6], the real data covers five modes, and we measure precision and recall when 1–10 of the modes are covered by a hypothetical generator that draws samples from the corresponding Gaussian distributions. In Figure 1b we see that our method yields the correct values for precision and recall in all cases: when not all modes are being generated, precision is perfect and recall measures the fraction of modes covered, and when extraneous modes are generated, recall remains perfect while precision measures the fraction of real vs. generated modes. Figure 1c illustrates that the method of Sajjadi et al. [6] performs similarly except for artifacts from $k$-means clustering.

## C  Analysis of truncation methods

Many generative methods employ some sort of truncation trick [5, 1, 4, 3] to allow trading variation for quality after the training, which is highly desirable when, e.g., showcasing uncurated results. However, quantitative evaluation of these tricks has proven difficult, and they are largely seen as an ad-hoc way to fine-tune the perceived quality for illustrative purposes. Using our metric, we can study these effects in a principled way.

StyleGAN is well suited for comparing different truncation strategies because it has an intermediate latent space $\mathcal{W}$ in addition to the input latent space $\mathcal{Z}$. We evaluate four primary strategies illustrated in Figure 2a: A) generating random latent vectors in $\mathcal{W}$ via the mapping network [3] and rejecting ones that are too far from their mean with respect to a fixed threshold, B) approximating the distribution of latent vectors with a multivariate Gaussian and rejecting the ones that correspond to a low probability density, C) clamping low-density latent vectors to the boundary of a higher-density region by finding their closest points on the corresponding hyperellipsoid [2], and D) interpolating all latent vectors linearly toward the mean [4, 3]. We also consider three secondary strategies: E) interpolating the latent vectors in $\mathcal{Z}$ instead of $\mathcal{W}$, F) truncating the latent vector distribution in $\mathcal{Z}$ along the coordinate axes [5, 1], and G) replacing a random subset of latent vectors with the mean of the distribution. As

Figure 2: (a) Our primary truncation strategies avoid sampling the extremal regions of StyleGAN's intermediate latent space. (b) Precision and recall for different amounts of truncation with FFHQ. (c) Using FID instead of recall to measure distribution quality. Note that the $x$-axis is flipped.

suggested by Karras et al. [3], we also tried applying truncation to only some of the layers, but this did not have a meaningful impact on the results.

Figure 2b shows the precision and recall of each strategy for different amounts of truncation. Strategies that operate in $\mathcal{Z}$ yield a clearly inferior tradeoff (E, F), confirming that the sampling density in $\mathcal{Z}$ is not a good predictor of image quality. Rejecting latent vectors by density (B) is superior to rejecting them by distance (A), corroborating the Gaussian approximation as a viable proxy for image quality. Clamping outliers (C) is considerably better than rejecting them, because it provides better coverage around the extremes of the distribution. Interpolation (D) appears very competitive with clamping, even though it ought to perform no better than rejection in terms of covering the extremes. The important difference, however, is that it affects all latent vectors equally — unlike the other strategies (A–C) that are only concerned with the outliers. As a result, it effectively increases the average density of the latent vectors, countering the reduced recall by artificially inflating precision. Random replacement (G) takes this to the extreme: removing a random subset of the latent vectors does not reduce the support of the distribution but inserting them back at the highest-density point increases the average quality.[2]

Our findings highlight that recall alone is not enough to judge the quality of the distribution — it only measures the extent. To illustrate the difference, we replace recall with FID in Figure 2c. Our other observations remain largely unchanged, but interpolation and random replacement (D, G) become considerably less desirable as we account for the differences in probability density. Clamping (C) becomes a clear winner in this comparison, because it effectively minimizes the Wasserstein-2 distance between the truncated distribution and the original one in $\mathcal{W}$. We have inspected the generated images visually and confirmed that clamping appears to generally yield the best tradeoff.

## D Quality of samples and interpolations

Figure 3 shows BigGAN-generated images for which the estimated realism score is very high or very low. Images with high realism score contain a clear object from the given class, whereas low-scoring images generally lack such object or the object is distorted in various ways. High and low quality images for each class were obtained from 1k generated samples.

Figure 4 demonstrates StyleGAN-generated images that have very high or very low realism score. Some variation in backgrounds, accessories, etc. is lost in high quality samples. We hypothesize that the generator could not realistically recreate these features, and thus they are not observed in high quality samples, whereas low quality samples often contain hats, microphones, occlusions, and

varying backgrounds that are challenging for the generator to model. High and low quality images were obtained from 1k generated samples.

Figure 5 presents further examples of high and low quality interpolations. High-quality interpolations consist of images with high perceptual quality and coherent background despite the endpoints being potentially quite different from each other. On the contrary, low-quality interpolations are usually significantly distorted and contain incoherent patterns in the image background.

## Footnotes

[2]Interestingly, random replacement (G) actually leads to a slight *increase* in recall. This is an artifact of our $k$-NN manifold approximation, which becomes increasingly conservative as the density of samples decreases.

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

(a) High-quality samples

(b) Low-quality samples

Figure 3: Examples of (a) high and (b) low quality BigGAN samples according to our realism scores.

(a) High-quality samples

(b) Low-quality samples

Figure 4: High (a) and low quality (b) StyleGAN samples according to our realism scores.

(a) High-quality interpolations

(b) Low-quality interpolations

Figure 5: Examples of (a) high and (b) low quality interpolations according to our realism scores.