[Reviews · NeurIPS 2019]

Reviewer 1



They try to estimate the manifold in the feature space of real images and the generated images separately. They do it calculating the pairwise Euclidean distances between all feature vectors in the set of real images and for each feature vector forming a hyper-sphere with radius equal to the distance to its kth nearest neighbor. Together, these hyperspheres define a volume in the feature space that serves as an estimate of the true manifold. Similarly, manifold of the generated images is estimated. Further, precision is estimates as the fraction of the generated images that lie in the manifold of the real images. And recall is estimated as the fraction of the real images that lie in the manifold of the generated images. Originality: I believe the proposed method of estimating precision-recall is not novel. Given its simplicity, it must have been used in some previous works although in a different context (not for high dimensional images). Quality: The paper is well written and results are clearly explained. Significance: Given their numerical experiments, the proposed method holds significance in evaluating the quality and variation of samples generated by GAN/

Reviewer 2



I found this submission really interesting and influential. It represents a great step forward compare to the prior work on recall/precision estimation of GANs. The authors show examples where previous way to estimate recall/precision fails to find any difference despite obviously varying quality of samples. The paper is very well-written and contains thorough experiments (for instance, one experiment semi-accidentally pushes StyleGAN state-of-the-art in FID). As a sidenote, the underlying idea of growing hyperspheres around data points reminded me about persistent homologies. It is a concept from computational topology which can be used to estimate manifold topology when the manifold is described by point cloud of samples. The authors may benefit from exploring this connection. For more details please see for example "Towards topological analysis of high-dimensional feature spaces" by Hubert Wagner and Paweł Dłotkoa (Computer Vision and Image Understanding, Volume 121, April 2014, Pages 21-26)

Reviewer 3



Originality: This paper uses similar intuition as [1]. Precision should represent the generated images captured by real images and the recall should represent the real images should be captured by generated images. Instead of using PR curve, the authors use two values definition as information retrieval metric and claim it is better by showing counterexample in StyleGAN with truncation trick. I think the main contribution is the empirical evaluations on large-scale GANs. They evaluated StyleGAN and BigGAN and show the tradeoff between precision and recall by controlling the truncation trick. They also proposed a method for evaluating a single image and demonstrate the effectiveness qualitatively. The authors demonstrate the potential of using these evaluation metrics to improve the training of GANs. Clarity: The paper is generally well-written and structured clearly. However, it would be better if the authors could write the precision and recall as a definition on probability measures and estimator on data samples. If I understood clearly, the definitions in (2) line 88 are actually estimator of precision defined as P(supp(Q)) and recall defined as Q(supp(P)). Quality: In theory, I do not think P(supp(Q)) and Q(supp(P)) are a good evaluation of distance of two probability distributions. One could easily come up with a counterexample: suppose two continuous probability distributions have same support set but different densities, e.g. two skewed gaussians. Then this method could not measure the performance since P(supp(Q)) and Q(supp(P)) are both 1. This method only works if two probability distributions are uniform. And I believe this is the reason why we need a PR curve like [1] instead of two values. In practice, this problem might be mitigated by carefully choosing the features in the pre-trained classifier and k. The authors mentioned these two choices in line 99 and line 104. It would be better if the authors could provide results for different choices of features and k, the method of choosing them and the analysis. If the number of samples, training data, or even generative model has changed, how should we adjust these settings to have consistent evaluations? As the authors explained at line 138,139,140, the bad clustering performance in the feature space is probably the reason why [1] failed, it would be critical to choose a good feature space and k. For experiment sections, it would be better if the authors can also provide results on some simple or synthetic datasets like [1] did, so we can have a fair comparison on these two methods. The authors claim (line 57, 58) that one of the drawbacks of [1] is that a curve representation is ambiguous. However, in Figure 4 (c), the authors actually use F-scores as proxies for precision and recall, which is proposed in [1]. I do not agree that curve representation is ambiguous as they can also be summarized by F-scores. They also claim another drawback of [1] is that the estimation algorithm is not practical. However, it is discussed in [2] that the definition can be extended to continuous distributions. A practical estimation algorithm based on binary classifier and binary hypothesis testing is also proposed in [2]. The motivation of using a PR curve/ROC curve and connection to binary hypothesis testing is discussed in [2, 3]. Significance: This paper proposes evaluation metrics of GANs and applies them on StyleGAN and BigGAN. The experiments are thoroughly conducted and discussed. I think it is a good contribution to ML community. [2] Simon, L., Webster, R., & Rabin, J. (2019). Revisiting Precision and Recall Definition for Generative Model Evaluation. arXiv preprint arXiv:1905.05441. [3] Lin, Z., Khetan, A., Fanti, G., & Oh, S. (2018). Pacgan: The power of two samples in generative adversarial networks. In Advances in Neural Information Processing Systems (pp. 1498-1507).

[Author Response · NeurIPS 2019]



| (a) Varying $k$, VGG-16 | (b) Varying $k$, Inception-v3 | (c) Varying $|\mathbf{\Phi}|$, VGG-16 |

Figure 1: (a) Precision (blue) and recall (orange) for several neighborhood sizes $k$. (b) Using Inception-v3 features instead of VGG-16 yields a substantially similar result. (c) Our metric behaves similarly to FID in terms of varying sample count.

| (a) Data | (b) Our method | (c) Sajjadi et al. [1] |

Figure 2: (a) Real data covers five modes (1–5) and the generated data is expanded, one mode at a time, to cover the real modes (1–5) and five extraneous modes (6–10). Both metrics were evaluated using 20k real and generated samples. (b) Results from our metric with $k = 3$. (c) Results from the method of Sajjadi et al. [1].

We thank the reviewers for their comments and remarks, and will gladly implement the suggested clarifications.

Reviewers 1 and 3 ask about different neighborhood sizes $k$, the number of samples $|\mathbf{\Phi}|$, and the choice of feature
space. Figure 1a illustrates the effect of varying $k$ in the setup used in Figure 4b of the submission (truncation sweep
in StyleGAN, VGG-16 features, 50k samples). In general, different $k$ yield consistent results and affect mainly the
saturation towards 0 or 1. Therefore, selecting $k$ is a tradeoff between under- or overestimating the manifolds. We
chose $k = 3$ for slight underestimation, as overestimation leads to quicker saturation of precision and consequently
makes it harder to measure differences between models. Figure 1b further shows that our metric is not sensitive to the
choice of feature space: extracting the features from *pool3* of Inception-v3 [3] instead of VGG-16 makes no qualitative
difference. Finally, Figure 1c shows that our metric behaves similarly to FID as the number of samples increases.

Reviewer 3 points out that our precision and recall do not measure the distance between generated and real distributions,
and gives a counterexample where two continuous probability distributions have the same support sets but different
densities. In this case our proposed metric would return perfect precision and recall scores, as it explicitly aims to
disregard the density of the target distribution, measuring only the probability that a sample drawn from one distribution
falls within the support of the other. FID remains an important tool for measuring distances between the distributions,
and we argue that precision, recall, and FID all have well-justified roles in evaluating generative models as they provide
complementary information about them.

Reviewer 3 further questions our claim that the curve representation in [1] is ambiguous. Making this claim was a result
of an unfortunate grammatical mistake in our paper on line 57. Our intent was to say that the choice in [1] to use curves
resulted *from* an ambiguity (that they discuss in Section 3.1), not that it came *with* any ambiguity. We apologize for
the error and will revise the text. We have no objections to summarizing the curves using $F_\beta$ scores as done in [1].
Furthermore, we thank the reviewer for bringing [2] to our attention, and will cite it as parallel work.

Finally, reviewer 3 suggested experimenting with simple or synthetic datasets similar to [1]. In Figure 2, we replicate
the mode dropping and invention experiment in [1], albeit with a 10-class 2D Gaussian mixture model instead of
CIFAR-10 images. As in [1], the real data covers five modes, and we measure precision and recall when 1–10 of the
modes are covered by a hypothetical generator that draws samples from the corresponding Gaussian distributions. In
Figure 2b we see that our method yields the correct values for precision and recall in all cases: when not all modes are
being generated, precision is perfect and recall measures the fraction of modes covered, and when extraneous modes are
generated, recall remains perfect while precision measures the fraction of real vs. generated modes. Figure 2c illustrates
that the method of Sajjadi et al. [1] performs similarly except for artifacts from $k$-means clustering. We agree that
including an experiment like this would strengthen the paper.

# References

[1] M. S. M. Sajjadi, O. Bachem, M. Lucic, O. Bousquet, and S. Gelly. Assessing generative models via precision and
recall. In *Proc. NIPS*, 2018.
[2] L. Simon, R. Webster, and J. Rabin. Revisiting precision and recall definition for generative model evaluation.
*CoRR*, abs/1905.05441, 2019.
[3] C. Szegedy, W. Liu, Y. Jia, P. Sermanet, S. E. Reed, D. Anguelov, D. Erhan, V. Vanhoucke, and A. Rabinovich.
Going deeper with convolutions. In *Proc. CVPR*, 2015.


[Meta-Review · NeurIPS 2019]

This paper proposes a new metric for mode collapse, which is a scalar value that can be read off from previously proposed measure of mode collapse in PacGAN. Precisely, in the mode collapse region, one can read the two points: (i) where the mode collapse region touches vertical axis ($\delta$-axis) and (ii) where the mode collapse r region touches \delta=1 line. Each one is exactly the same as P_r(support{P_g}) and P_g(support{P_r}) that defend the proposed scalar valued mode collapse measure. This should be explained precisely in the paper, as (i) PacGAN introduced a proper mathematical notion of mode collapse earlier, (ii) the mode collapse region strictly generalizes the proposed metric (iii) mode collapse regions is the foundation of understanding mode collapse theoretically. A new estimator based on nearest neighbor distances are proposed, with extensive numerical validation of the proposed metric.